# Nutritional and Educational Intervention to Recover a Healthy Eating Pattern Reducing Clinical Ileostomy-Related Complications

**DOI:** 10.3390/nu14163431

**Published:** 2022-08-20

**Authors:** Antonio Fernández-Gálvez, Sebastián Rivera, María del Carmen Durán Ventura, Rubén Morilla Romero de la Osa

**Affiliations:** 1General and Digestive Surgery Department, University Hospital Virgen del Rocio, 410013 Seville, Spain; 2Department Nursing, Faculty of Nursing, Physiotherapy and Podiatry, University of Seville, 410013 Seville, Spain; 3Institute of Biomedicine of Seville, University Hospital Virgen del Rocio, CSIC, University of Seville, 410013 Seville, Spain; 4Centro de Investigación Biomédica en Red de Epidemiología y Salud Pública (CIBERESP), 28029 Madrid, Spain

**Keywords:** ileostomy, eating pattern, self-management, organism hydration status, body weight, hospital readmission

## Abstract

The aim of this study was to evaluate a diet intervention implemented by our hospital in order to determinate its capacity to improve the eating pattern of patients with an ileostomy, facilitating the implementation new eating-related behaviors, reducing doubt and dissatisfaction and other complications. The study was conducted with a quasi-experimental design in a tertiary level hospital. The elaboration and implementation of a nutritional intervention consisting of a Mediterranean-diet-based set of menus duly modified that was reinforced by specific counseling at the reintroduction of oral diet, hospital discharge and first follow-up appointment. Descriptive, bivariate and multivariate analyses were performed. The protocol was approved by the competent Ethics Committee. The patients of the intervention group considered that the diet facilitated eating five or more meals a day and diminished doubt and concerns related to eating pattern. Most patients (86%) had a favorable experience regarding weight recovery and a significant reduction of all-cause readmissions and readmission with dehydration (*p* = 0.015 and *p* < 0.001, respectively). The intervention helped an effective self-management of eating pattern by patients who had a physical improvement related to hydration status, which, together with an improvement in weight regain, decreased the probability of readmissions.

## 1. Introduction

In the management of colorectal cancer (CRC) or inflammatory bowel disease (IBD), it may be necessary to perform an ileostomy, which is a skin exteriorization from an ileal segment to eliminate feces either temporarily or permanently [1].

On average, 50% of the cases present clinical complications, however, this percentage can rise to 96% during the first three post-operative weeks where hydroelectrolytic alterations account for 20–29% [2]. In this period, up to 16% of these patients will present high fecal outputs through the stoma (>2000 mL/24 h), which supposes higher risk of dehydration, electrolyte imbalance and malnutrition [2]. Thus, readmission 60 days after discharge is common in these patients [3].

Factors such as the surgery, the underlying pathology and other treatment options are related to the nutritional and hydroelectric changes reported. Micronutrient deficit, malabsorption of bile fats and salts, caloric malnutrition and hydroelectric and weight losses are associated with the surgical procedure [4,5]. Reduced intake and metabolic changes, as well as higher energy expenditure, are described in CRC [6,7]; energy and protein malnutrition with deficit of essential micronutrients is common in IBD [8], whereas chemotherapy and radiotherapy produce symptoms related to a higher risk of malnutrition [9].

Early physiological enteral nutrition must be the first option since it eases the process of intestinal adaptation, preserves the intestinal flora and the enteric immune system, presents fewer complications and is cost effective when compared with other options [10]. In general, these patients will be able to follow a normal diet avoiding certain food products and introducing some changes in their habits aimed at preventing stoma obstruction, dehydration and gastrointestinal discomfort or food intolerance [11].

The physical and psychosocial implications associated with the implantation of the stoma, together with the absence of sound diet recommendations, leads many patients to adopt heterogeneous strategies of doubtful value based on their experiences or on myths about the properties of certain food products, which exert a negative impact on their nutritional state [12]. They frequently report receiving scarce or no information, and sometimes contradictory information about diet guidelines, which generates anxiety, confusion and/or frustration in them, thus increasing dissatisfaction with the care received [13]. In addition, providing them with strong diet recommendations to which they can adhere would be fundamental to avoid future complications and readmissions [14].

In this sense, an inadequate health education limits their ability to assume responsibility for self-care in an effective manner [15,16]. Self-efficacy has shown to be an indispensable factor in habit change related to motivation, adherence to treatment and, ultimately, with the improvement of outcomes [17].

Based on this, the aim of this study was to evaluate a diet intervention implemented by our hospital in order to determinate its capacity to improve the eating patterns of patients with an ileostomy, facilitating the implementation of the diet guidelines, minimizing the doubts in relation to their diet, and reducing their levels of concern, dissatisfaction and complications. This type of intervention could improve the patients’ autonomy in the management of their disease, which would suppose an advancement of great interest in clinical practice.

## 2. Materials and Methods

### 2.1. Setting

The study was conducted in the General Digestive Surgery Unit of a tertiary level hospital where 166 ileostomies were performed in adults from 2019 to 2020. To improve the care provided to patients with ileostomy, this unit, working collaboratively with the Endocrinology and Nutrition Unit, developed a nutritional intervention consisting of a health education session and a set of menus as support material home care (Appendix B) for patients, who previously only received brief written instructions at discharge (Appendix A).

### 2.2. Design

A quasi-experimental design of unpaired samples was carried out. To ensure the methodological quality of the study, recommendations from the Methodological Index for Non-Randomized Studies (MINORS) [18] and from the Joanna Briggs Institute’s Critical Appraisal Checklist for Quasi-Experimental Studies (non-randomized studies) were followed [19].

### 2.3. Sample Size

Sample size was calculated by maximizing the parameters for the comparisons of proportions and means (for outcomes measured with a 5-point Likert scale). The calculation that rendered the largest sample size was chosen. The ability to detect a 1-point difference in the Likert scale with a 5-point variance was pre-fixed, assuming a 10% loss (estimation of refusals to sign the informed consent). For a bilateral test with 90% power and 95% statistical power, the calculation indicated 117 patients.

### 2.4. Intervention

The elaboration and implementation of this intervention included endocrinologists, nutrition technicians and stoma therapists. A specific oral diet with a Mediterranean pattern was designed due to its widely proved beneficial effects to health, and for being the characteristic eating pattern of this region, easing adherence. Liquid, initial, soft (taken during hospitalization) and basal (taken at home) variants were elaborated to allow its progressive and successful introduction. For each variant, autumn/winter and spring/summer models were elaborated, allowing the use of seasonal food products in its preparation. For basal diets, 14 full menus were designed for lunch and dinner, as well as examples for breakfast and teatime, which were handed in to the patients after discharge (Appendix B).

Furthermore, patients received an educational session three times in different moments (reintroduction of diet after surgery, at discharge and during first visit after discharge). All lasted around 30–35 min where the stoma therapist explained key information on nutrition and eating habits to avoid complications during the first session. The key messages were summarized and provided in writing to the patient in the first session (Appendix B) and served as a follow-up script for the remaining sessions when the stoma therapist received feedback on the habits implemented and difficulties or complications associated with the eating dimension, providing solutions and reinforcing positive habits.

### 2.5. Procedures

The design of the menus and the health education session, as well as their approval by the hospital management, lasted from October 2017 to April 2018. The implementation of the intervention began in April 2018, when the prospective recruitment of patients for the intervention group began, and lasted until July 2020. The patients were recruited after surgery according to inclusion criteria (adults with ileostomy who signed the informed consent form, followed-up in our unit and who stated their intention to follow the diet proposed). Subjects with sensory/cognitive constraints and non-Spanish speakers were excluded due to the risk of misunderstanding the indications.

The control group was recruited retrospectively during their medical review visits between January and July 2018, excluding patients who underwent surgery before 2016 to minimize the memory bias.

A 23-item survey was elaborated, where sociodemographic (age, gender, marital status, schooling level, occupation, family situation), clinical (type of ileostomy, centimeters of ileum removed, etiology, chemotherapy treatment, incidence of gastrointestinal symptoms in case of chemotherapy treatment, self-care level) and anthropometric (weight and BMI) variables were measured. Due to the absence of a validated scale to determine the self-care level in these patients, this is usually determined by the stoma therapist through observation and interviews with the patients, who were assigned autonomous, semi-dependent or dependent levels.

### 2.6. Outcomes

To assess outcomes related to eating pattern management a 5-point Likert scale was used to determine the difficulty to implement the diet recommendations and the usefulness of the intervention by the patients. Additionally, onset of gastrointestinal problems related to the diet, compliance of the diet guidelines received, emergence of doubts in relation to the food products to eat and to their preparation, and if they considered that the diet was adequate were collected as dichotomous (yes/no) questions. The number of meals eaten per day and evolution of weight were also collected. The control group patients were asked if they recalled having gained, lost or maintained weight after 1–3 months of the ileostomy. The intervention group was followed up obtaining BMI values at the time of the surgery (baseline), weight at discharge, and at the first and second scheduled appointments (7–14 days and 30–60 days after discharge, respectively). Readmissions at 60 days after discharge, including dehydration among causes, were collected from clinical digital records from our emergency department.

### 2.7. Data Analysis

Data were analyzed in the strictest confidentiality with IBM SPSS software v22 (IBM, Chicago, USA). A descriptive analysis by absolute and relative frequencies and median and interquartile range was applied as appropriate. Pearson’s chi-square test was used to evaluate the association in qualitative variables. Wilcoxon signed-rank test and Kruskal–Wallis test were performed for comparison of means checking normality tests beforehand. Finally, multivariate models were implemented to elucidate possible confounding bias. Statistical significance was defined as obtaining a *p*-value under 0.05.

### 2.8. Ethical Aspects

The Declaration of Helsinki and Guides for a Good Clinical Practice were taken into account to conduct this study. Protocol was approved by the competent Ethics Committee. The patients were informed about the study objectives and dynamics prior to their inclusion. This information was offered together with the informed consent document on the follow-up visits for the control group patients, and after the surgery in the intervention group patients.

## 3. Results

The number of patients included was 253, with 117 in the control group (71% of all the patients seen during the inclusion period) and 136 in the intervention group, where, initially, 164 met the inclusion criteria, but there were 28 losses (5 refusals to participate, 2 problems with language, 3 relevant missing data, 6 high-debit ileostomy, 9 unfavorable life prognosis and 3 deaths).

Age ranged between 18–89 years old, distributed with a mean and standard deviation of 58.5 ± 17.5 and 59.3 ± 15.6 in the control and intervention groups, respectively, without differences between them (*p* = 0.854). Except for the educational level (*p* = 0.042), no statistically significant differences were found between the groups for the sociodemographic characteristics (Table 1).

Table 2 shows the descriptive analysis of the clinical, anthropometric and outcome variables. No significant differences related to etiology, type of ileostomy or autonomy in care were observed between groups. There was difference for the “centimeters of ileum removed”, whose mean and standard deviation were 20.5 ± 43.1 and 10.7 ± 18.6 cm in the control and intervention groups (*p* = 0.016), respectively.

The most frequent etiology was CRC (60.9%) followed by inflammatory disease (27.3%) and other causes (11.8%) where different types of trauma were included. However, no differences were observed between groups regarding etiology. On the other hand, statistically significant differences were found for all outcomes, always in favor of the intervention group. Patients in the control group were asked if they thought that such an intervention would have been useful for a better management of their eating pattern. Among them, 109/117 (93.1%) assessed possible usefulness with four and five points (Likert scale). These points were assigned by 133/136 (98%) of the intervention group patients. It was not possible to assess statistical differences due to that, in the control group, the question explored the need, and in the intervention group, it was confirmatory of usefulness.

In the logistic regression models the intervention was identified as a risk factor to consider that the diet was adequate, and as a protective factor for eating five meals or more a day, being concerned when preparing the meals and having doubts in relation to the diet. A linear regression model showed that the intervention maintains an inverse relationship with the difficulty to implement the recommendations (Table 3).

Weight gain and weight loss was reported by 43 (36.75%) and 69 (58.97%) control patients, respectively, leaving only 5 (4.27%) with a stable weight after the surgery. In the intervention group (Table 4) significant differences were observed at discharge and follow-up appointments 1 and 2, although this only involved significant differences in the weight loss percentage in the last assessment coinciding with significant difference in days from baseline to this timepoint. Among them, 92% attended the first scheduled review appointment at 10.5 ± 4.8 days and 96% attended the second appointment at 42.5 ± 10.9 days (mean and SD).

Regarding readmissions at 60 days after discharge, patients in the control group had a readmission rate of 29.7%, with the specific rate of readmission due to dehydration being 17.8%. This values for intervention group were 16.2% and 4.4%, respectively. A statistically significant reduction between groups was observed for theses outcomes (*p* = 0.015 and *p* < 0.001, respectively). Odd ratios (CI95%) for intervention regarding the control group were 0.46 (0.24–0.87) for total readmissions and 0.21 (0.07–0.56) for specific readmission due to dehydration.

## 4. Discussion

This is a quasi-experimental study that has assessed a nutritional intervention in patients with an ileostomy. Usually, these patients simply receive a list of non-recommendable food products and culinary techniques, previous to their nutritional assessment [12,17,20,21,22,23]. However, we have not found papers that offer an instrument to mitigate the patient’s uncertainties during the elaboration of meals and improve the self-management of their eating pattern while inducing improvements related to weight and fluid balance.

### 4.1. Patients’ Profiles

It was similar (57.7% men and 59 ± 16.5 years old) to other Western countries with the predominance of CRC patients (55.6% in the control group and 65.4% in the intervention group, *p* > 0.05) [24,25].

Educational level was lower in the control group, which could affect the understanding of the recommendations. Although 52% of the subjects were considered as autonomous patients in self-care, only two individuals stated living alone and without the support of family members, implying that most of them would have their care needs covered.

The main etiology was malignant pathology (55.6% in the control group and 65.4% in the intervention group), as previously reported [26]. Although there was no statistical difference in the cancer prevalence between the groups, it was verified by the number of patients who underwent chemotherapy (*p* = 0.049); however, the rates of gastrointestinal symptoms were similar, excluding this as a possible bias to improve the eating pattern.

### 4.2. Outcomes

The multivariate models showed that the intervention is a highly protective factor regarding the patients’ concern at the time of preparing the meals (OR: 0.05, 95%CI: 0.02–0.12), and the doubts related to the food products (OR: 0.08, 95%CI: 0.04–0.14). This is a positive result since patients with ileostomy feel anxiety, confusion and frustration in relation to how to address diet [13]. Thus, this could lead them to adapt recommendations to their lifestyles and preferences on their own [27], increasing the risk of doing so inadequately. An inverse relationship was observed between belonging to intervention group and having difficulties in implementing the recommendations. These benefits are corroborated with the results observed in Models 1 and 3 (Table 3). In addition, our research significantly reduced the proportion of patients with gastrointestinal problems related to diet (OR:0.06, 95%CI:0.03–0.37).

Reducing the amount of food eaten in each meal and increasing the number of meals per day is beneficial for these patients [28,29]. Thus, Mukhopadhyay recommended an oral diet with six–eight meals/day with a reduction in the amounts eaten, leading all patients to recover their pre-surgery weight within 3 months [10].

Intestinal adaptation favors a partial recovering of intestinal function. This process is conditioned by the presence of food and secretions and begins 2–3 months after surgery [5]. Observational studies including patients, showed weight gain with respect to their pre-surgery weight 12 and 8 weeks after the procedure [10,21]. Likewise, in a retrospective study 13.3% of the patients lost weight after surgery and 68% presented normal weight at reconstruction surgery [22]. Vasilopoulos observed severe weight loss (>3 kg) in 53.8% of patients 3 weeks after the surgery [17]. At 6 weeks, this percentage rose to 70% when severe weight loss was defined as that higher than 7.5% of the pre-surgery weight [23]. Our results show a weight loss >7.5% regarding pre-surgery weight in 28.7% of the intervention group patients at 12.2 ± 8.3 days (discharge). This rose to 35.3% at 22.4 ± 10.8 days (first appointment), meaning an improvement compared with those previously reported. At 54.8 ± 13.9 days (second appointment) this ratio decreased to 25%. These severe weight losses could be a consequence of the post-surgical anatomical and functional loss and the absence of the intestinal adaptation process.

Other authors used the day of ileostomy as baseline to assesses the weight gain. However, we adopted the weight at discharge because it involved the self-management of the eating pattern, allowing observation of the results of the diets and nutritional guidelines implemented. All the weight assessments were within 90 days after ileostomy, with a mean and standard deviation of 54.8 ± 13.9 days. In the intervention group, 95 (69.85%) patients presented higher weight at the second appointment (42.5 ± 10.9 days).

The multivariate analysis shows that the weight loss differences were associated with pre-surgical BMI (higher in overweight and obese patients) rather than differences in days to attend the appointment. This could be prompted by the previous need to lose weight for the patients, who took advantage to obtain a healthy weight according to the clinical recommendations. 

These results evidence, as previously, that early and maintained implementation of an oral diet is successful in recovering a patient’s weight and/or to steer their BMI into a healthy range [10,21].

The intervention involved a significant reduction in readmission rate for all causes and due to dehydration (*p* = 0.015 and *p* < 0.001, respectively). A recent systematic review that included 27,089 patients showed that the global incidence of 30- and 60-day readmission with dehydration were 5.0% (range 2.1–13.2%) and 10.3% (range 7.3–14.1%), respectively [30]. In our study, the 60-day readmission with dehydration rates were 4.4% for intervention and 17.8% for the control group. The rate of readmission with dehydration in our intervention group was significant lower than three studies (marked in bold) that reported similar results to our control group. However, our control group showed a more increased rate than most of the studies included, but our intervention reduced the rate so that no significant differences were observed for the group that received it (Table 5). These improvements could be related to the increase in the number of intakes observed in the intervention group, since these could also be accompanied by a higher water intake.

### 4.3. Strengths/Limitations

An experimental design to compare the intervention vs. standard care was not possible because the hospital management ordered the intervention to be applied to all patients, considering that it was beneficial enough to exclude it in a group of subjects. For this reason, we were forced to retrospectively choose a control group prior to the implementation of the protocol in the service.

Although significant differences were observed regarding the number of cm of ileus removed (greater in the control group), they were a number small enough to have clinical repercussions regarding the nutrition and hydration of the patients, since good tolerance of the loss has been described up to the middle of the small intestine (usual length from 6–8 m) [5]; none of the included patients lost lengths higher than 350 cm.

Our design represents an advantage over most previous studies, which were observational and did not contemplate a multivariate analysis to control the confusion bias derived from obtaining differences between groups such as “cm removed” or “educational level”. Moreover, our sample size is larger than those previously reported, and this is the reason why we consider that there was high statistical power.

The lower rate of readmissions to the emergency room due to dehydration could be attributed to the fact that the intervention group was discharged during the COVID-19 pandemic and patients were afraid to go to the hospital; however, in our country during the confinement primary care services delivered telehealth care only. In addition, dehydration in this type of patient is severe and requires intravenous fluid therapy and being admitted to the hospital, so we ruled out that the patients would stay at home.

Finally, our intervention is complex because through the menus and recommendations we work on the nutrients provided that are of interest to these patients (fiber or fat), types of food and eating-related behaviors, making it difficult to identify a specific factor responsible for the improvement of results. However, this is common in the clinical context, where, unlike a laboratory, not only one variable is controlled, but many with the intention of offering comprehensive care to the patient.

## 5. Conclusions

This study is pioneering in proving the effectiveness of a nutritional intervention consisting of a Mediterranean-diet-based set of menus duly modified to allow for a progressive and effective self-management of the eating pattern of patients with an ileostomy. This was complemented with recommendations provided in an initial educational session and reinforced in two subsequent sessions; in addition, a brief written guide on these recommendations was added as documentation for home.

The success of this complex and multiapproach intervention has been evidenced through physical improvements observed in weight recovery, an improved hydration and a lower proportion of patients with gastrointestinal problems, behavioral improvements that were reflected in an increase in the number of intakes, decreased number of doubts regarding food selection/preparation, and readmission rate. The patients’ satisfaction level was reflected through the positive assessment of the usefulness of the intervention and the consideration of having followed an adequate diet.

## Figures and Tables

**Table 1 nutrients-14-03431-t001:** Sociodemographic features.

Variable	Total n (%)	Control n (%)	Intervention n (%)	*p* Value	OR (CI95%)
Sex, male	146 (57.7)	68 (58.1)	78 (57.4)	0.902	0.97 (0.57–1.65)
Marital Status				0.933	
Single	41 (16.2)	19 (16.2)	22 (16.2)		ref
Married/Partner	179 (70.8)	82 (70.1)	97 (71.3)		1.02 (0.47–2.13)
Divorced/Separated	12 (4.7)	5 (4.3)	7(5.1)		1.2 (0.28–5.67)
Widowed	21 (8.3)	11 (9.4)	10 (7.4)		0.79 (0.24–2.56)
Education				0.042	
None	28 (11.1)	16 (13.7)	12 (8.8)		ref
Primary	113 (44.7)	59 (50.4)	54 (39,7)		1.2 (0.49–3.10)
Secondary	78 (30.8)	26 (22.2)	52 (38.2)		2.6 (1.00–7.10)
Higher	34 (13.4)	16 (13.7)	18 (13.2)		1.5 (0.49–4.63)
Occupation				NA	
Student	6 (2.4)	1 (0.9)	5 (3.7)		ref
Employed	63 (24.9)	20 (17.1)	43 (31.6)		0.43 (0.08–4.26)
Unemployed	41 (16.2)	23 (19.7)	18 (13.2)		0.16 (0.03–1.64)
Retired	143 (56.5)	73 (62.4)	70 (51.5)		0.19 (0.04–1.19)
Whom do you live with?				0.299	
Alone	28 (11.1)	16 (13.7)	12 (8.8)		ref
Couple	181 (71.5)	84 (71.8)	97 (71.3)		1.54 (0.64–3.78)
Son/Daughter	15 (5.9)	4 (3.4)	11 (8.1)		3.55 (0.8–19.27)
Parents	29 (11.5)	13 (11.1)	16 (11.8)		1.63 (0.51–5.32)
With Familial/Social Support *	26 (92.9)	14 (87.5)	12 (100)	0.492	NA

* This was only answered by patients who reported living alone.

**Table 2 nutrients-14-03431-t002:** Clinic, anthropometric and satisfaction variables.

Variable	Total n (%)	Control n (%)	Intervention n (%)	*p* Value	OR (CI95%)
Type of ileostomy, temporary	238 (94.1)	110 (94.0)	128 (94.1)	0.973	0.98 (0.30–3.29)
Self-care autonomy				0.321	
Autonomy	133 (52.6)	61 (52.1)	72 (52.9)		reference
Semi-independent	95 (37.5)	41 (35.0)	54 (39.7)		1.16 (0.65–1.96)
Dependent	25 (9.9)	15 (12.8)	10 (7.4)		0.57 (0.21–1.46)
Ileostomy aetiology				0.101	
Cancer	154 (60.9)	65 (55.6)	89 (65.4)		reference
Inflammatory pathology	69 (27.3)	33 (28.2)	36 (26.1)		1.25 (0.67–2.32)
Other	30 (11.8)	19 (16.2)	11 (8.1)		2.35 (0.99–5.88)
If cancer CT Trt? Yes *	44 (28.6))	24 (36.9)	20 (22.5)	0.023	2.34 (1.04–5.33)
If CT Trt GI symptoms? Yes *	21 (47.7)	11 (45.8)	10 (50.0)	0.783	0.85 (0.22–3.25)
GI problems related with eating, yes	111 (43.9)	89 (76.1)	22 (16.2)	<0.001	0.06 (0.03–0.12)
If GI problems, earlier appointment, yes *	9 (8.1)	8 (9.0)	1 (4.5)	0.988	0.99 (0.18–10.3)
Follow diet guidelines, yes	243 (96.0)	108 (92.3)	135 (99.3)	0.005	0.09 (0.002–0.66)
Doubts about feeding, yes	117 (46.2)	88 (75.2)	29 (21.3)	<0.001	0.09 (0.05–0.17)
Concern about meal preparation, yes	62(24.5)	54 (46.2)	8 (5.9)	<0.001	0.07 (0.03–0.17)
Adequate diet, yes	207 (81.8)	84 (71.8)	123 (90.0)	<0.001	3.68 (1.67–8.13)
Number of daily meals				<0.001	
<3	1 (0.004)	1 (0.01)	0 (0.0)		NA
3	23 (9.1)	23 (19.7)	0 (0.0)		NA
4	72 (28.5)	56 (47.9)	16 (11.8)		0.13 (0.06–0.27)
5	100 (39.5)	31 (26.5)	69 (50.7)		reference
≥6	57 (22.5)	6 (5.1)	51 (37.5)		3.79 (1.42–11.96)
Difficulty to implement dietary recommendations, Likert 1–5				<0.001	
1	86 (34.0)	13 (11.1)	73 (53.7)		Reference
2	76 (30.0)	34 (29.1)	42 (30.9)		0.22 (0.10–0.49)
3	58 (22.9)	42 (35.9)	16 (11.8)		0.07 (0.03–0.17)
4	22 (8.7)	18 (15.4)	4 (2.9)		0.04 (0.01–0.15)
5	11 (4.3)	10 (8.5)	1 (0.7)		0.02 (0.004–0.15)
Utility of new diet, Likert 1–5				Non	comparable
3	11 (4.3)	8 (6.8)	3 (2.2)		
4	95 (37.5)	44 (37.6)	51 (37.5)		
5	147 (58.1)	65 (55.6)	82 (60.3)		
Ileostomy (cm removed), mean (sd)		20.5 ± 43.1	10.7 ± 18.6	0.016	–

CT trt: chemotherapy treatment, GI: gastrointestinal, OR: odds ratio, CI95%: confidence interval 95%, NA: not available because data are insufficient for this analysis, cm: centimeters, sd: standard deviation. * These frequencies and percentages were calculated based on the number of patients included in the categories or variables to which they are conditioned.

**Table 3 nutrients-14-03431-t003:** Multivariate models for outcomes.

Multivariante Regression Logistic Models					
	Model 1 ^AIC: 228.96^	Model 2 ^AIC: 217.77^	Model 3 ^AIC: 239.99^	Model 4 ^AIC: 271.74^	Model 5 ^AIC:71.74^
	aOR (CI95%)	aOR (CI95%)	aOR (CI95%)	aOR (CI95%)	aOR (CI95%)
Group (Intervention)	3.72 *** (1.88–7.71)	0.05 *** (0.02–0.12)	0.05 *** (0.02–0.10)	0.08 *** (0.04–0.14)	
Education		1.88 ** (1.27–2.86)	0.62 * (0.41–0.91)	1.52 * (1.06–2.24)	
Ileost. (Temporary)		0.26 * (0.07–0.94)			
Marital St (Single)			1.02 (0.40–2.56)		
Marital St (Div/Sep)			9.68 * (2.22–46.78)		
Marital St (Widowed)			1.99 (0.64–6.45)		
Self-Care Autonomy				1.52 * (1.06–2.24)	8.55 * (1.78–154.12)
GI probl r/w eating					0.07 * (0.004–0.37)
Multivariante lineal regression model (Difficulty to implement dietary recommendations)					
	coef	*p*-value (model)		Adjusted R2	
β0	2.58 (***)	(***)		0.26	
Group (Intervention)	−0.97 (***)				
GI probl r/w eating	0.31 (*)				

Model 1: adequate diet, Model 2: concern about meals preparation, Model 3: number of daily meals ≥5, Model 4: doubts, Model 5: follow diet guidelines, Ileost: ileostomy, St: status, Div/Sep: divorced/separate, GI probl r/w eating: gastrointestinal problems related with eating, coef: coefficient, OR: odds ratio, CI: confident interval, * *p* < 0.05; ** *p* < 0.01; *** *p* < 0.001.

**Table 4 nutrients-14-03431-t004:** Analysis of the weight gain.

Baseline		Discharge			Follow-Up App 1			Follow-Up App 2		
BMI cat. (n)	Weight $\bar{x}$ ± SD	Weight $\bar{x}$ ± SD	WG $\bar{x}$ ± SD (avg %)	Days	Weight $\bar{x}$ ± SD	WG $\bar{x}$ ± SD (avg %)	Days $\bar{x}$ ± SD	Weight $\bar{x}$ ± SD	WG $\bar{x}$ ± SD (avg %)	Days $\bar{x}$ ± SD
Inadequate (5)	17.9 ± 0.6	68.3 ± 12.2	−2.7 ± 1.5 (−5.6)	10.6 ± 6.5	69.2 ± 11.8	−2.8 ± 1.5 (−5.8)	19.4 ± 8.1	48.6 ± 5.5	−0.3 ± 2.1 (−0.8)	70.0 ± 22.5
Healthy (53)	22.3 ± 1.8	66.7 ± 13.7	−3.6 ± 3.4 (−5.6)	12.1 ± 7.9	65.8 ± 14.1	−3.6 ± 4.0 (−5.9)	23.0 ± 13.3	62.4 ± 8.2	−0.8 ± 4.9 (−0.9)	49.8 ± 12.9
Overweight (51)	27.5 ± 1.4	67.3 ± 12.7	−4.3 ± 3.1 (−5.5)	13.4 ± 10.5	67.4 ± 12.7	−5.1 ± 3.8 (−6.8)	22.1 ± 10.0	72.2 ± 10.7	−3.2 ± 3.9 (−4.3)	56.5 ± 14.1
Obese (27)	32.9 ± 2.3	70.9 ± 14.8	−5.8 ± 3.6 (−6.6)	10.1 ± 5.5	70.4 ± 13.8	−6.8 ± 4.6 (−7.8)	22.3 ± 7.4	82.2 ± 14.0	−6.3 ± 6.2 (−7.2)	58.5 ± 9.8
*p*-Value (KW test)			0.04 (0.719)	0.7862		0.02 (0.782)	0.7862		<0.001(0.007)	<0.001
**Multivariante regression lineal models**										
	**WG in Follow-Up App 2** ** *p* ** **-Value (Model): <0.001**			**Avg % of WG in Follow-Up App 2** ** *p* ** **-Value (Model): <0.001**						
Independent Variables	**Coefficient**	** *p* ** **-Value**	**Adjusted** **R Square**	**Coefficient**	** *p* ** **-Value**	**Adjusted** **R Square**				
β_0_	10.68	<0.001	0.2026	12.56	<0.001	0.1538				
IMC category	−0.52	0.748		−0.67	<0.001					
days	0.004	0.889		0.013	0.748					

BMI cat.: body mass index categories, KW test: Kruskall–Wallis test, App: appointment, $\bar{x}$: mean, SD: standard deviation, WG: weigh gain, Avg: average.

**Table 5 nutrients-14-03431-t005:** Comparison between rates of readmission with dehydration in intervention and control groups with previous studies.

Study *	Rate Reported *	Intervention Group (Rate: 4.4%) OR (CI95%)	Note	Control Group (Rate: 17.8%) OR (CI95%)	Note
Alqahtani et al. 2020	2.1	2.2 (0.08–4.9)	nsd	9.7 (5.7–15.9)	ssi
**Charak et al. 2018**	**14.1**	**0.28 (0.08–0.82)**	**ssr**	**1.31 (0.6–2.98)**	**nsd**
Chen et al. 2018	2.9	1.5 (0.54–3.46)	nsd	7.14 (4.16–11.78)	ssi
**Fish et al. 2017**	**11.5**	**0.35 (0.12–0.86)**	**ssr**	**1.66 (0.90–2.99)**	**nsd**
Glasgow et al. 2014	13.2	0.31 (0.08–1.12)	nsd	1.42 (0.53–4.25)	nsd
**Justiniano et al. 2018**	**11.1**	**0.37 (0.12–0.94)**	**ssr**	**1.17 (0.89–3.33)**	**nsd**
Li et al. 2017	3	1.49 (0.57–3.35)	nsd	6.98 (3.74–12.77)	ssi
McKenna et al. 2017	4.6	0.95 (0.33–2.34)	nsd	4.47 (2.41–8.09)	ssi
Messari et al. 2012	7.3	0.59 (0.20–1.42)	nsd	2.75 (1.48–4.96)	ssi
Paquette et al. 2013	7.5	0.57 (0.18–1.61)	nsd	2.68 (1.25–5.85)	ssi

OR: odd ratio, CI95%: confidence interval 95%, nsd: no significant differences, ssr: statistically significant reduction, ssi: statistically significant increase. * Information taken from Liu et al., 2021 [30].

## Data Availability

Not applicable.

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
