# Peer review of "Nutritional and Educational Intervention to Recover a Healthy Eating Pattern Reducing Clinical Ileostomy-Related Complications"

_nutrients, 2022, doi:10.3390/nu14163431_

Round 1

Reviewer 1 Report

Table 5 should present previously the strengths and limitations.

In general, the paper was very clear and the results are very important for the reinforcement the Mediterranean diet, with clinical applications.

Author Response

Dear reviewer 1,

we appreciate the trouble you have taken to review our work. Based on your comment, which we see as correct, we have placed table 5 before the strengths/limitations section.

best regards

Reviewer 2 Report

The authors reported a new diet intervention for patients with an ileostomy to prevent postoperative dehydration. They used this intervention to evaluate postoperative weight changes and satisfaction with the intervention. The results showed patients in the intervention group got improvements in weight recovery, improved hydration, and lower incidence of complications.  Thus, they claim that their new nutritional intervention is feasible for patients with an ileostomy.

Overall, I believe that this manuscript can be useful insight for the postoperative management of patients with an ileostomy. 

The authors claimed that the most of patients had a favorable experience regarding weight recovery and a significant reduction of all-cause readmissions and readmission with 24 dehydration. However, in their comparison, there was a difference for the centimeters of ileum removed.

Although they mentioned this finding in the limitation, I could not convince their explanation. Was there a difference in stool volume or stool morphology between the two groups? I think that data with estimated lengths of residual small intestine would be more convincing and show the usefulness of their Intervention. Then, Table 2 should show the estimated remaining small intestine length or resected intestinal length. Even if you cannot show convincing data, you should show current data with its p-value.

Their Intervention includes a relatively high-calorie, high-fat diet. Postoperative patients are expected to have a lower appetite. How much could the patients eat? Is there possible improvement in them? How about, in your next study, do you need to consider reducing calories and fat content more?

The authors recommend chewing food very well. Do you give consideration to the form of the food, such as feeding finely chopped meals?

Author Response

dear reviewer,
We are grateful for accepting the review of our article and for the contributions it makes to us, which will undoubtedly open the scientific debate around our study.
Attached we send you a document where you will find the answers to your comments and we hope you find our answers interesting

best regards
